# Angular-Resolved Thomson Parabola Spectrometer for Laser-Driven Ion Accelerators

**DOI:** 10.3390/s22093239

**Published:** 2022-04-22

**Authors:** Carlos Salgado-López, Jon Imanol Apiñaniz, José Luis Henares, José Antonio Pérez-Hernández, Diego de Luis, Luca Volpe, Giancarlo Gatti

**Affiliations:** Centro de Láseres Pulsados (CLPU), Edificio M5, Parque Científico USAL, C/Adaja, 8, 37185 Villamayor, Salamanca, Spain; japinaniz@clpu.es (J.I.A.); jlhenares@clpu.es (J.L.H.); japerez@clpu.es (J.A.P.-H.); ddeluis@clpu.es (D.d.L.); lvolpe@clpu.es (L.V.); ggatti@clpu.es (G.G.)

**Keywords:** plama diagnostics, charged-particle spectroscopy, ion beams, instrumentation

## Abstract

This article reports the development, construction, and experimental test of an angle-resolved Thomson parabola (TP) spectrometer for laser-accelerated multi-MeV ion beams in order to distinguish between ionic species with different charge-to-mass ratio. High repetition rate (HHR) compatibility is guaranteed by the use of a microchannel plate (MCP) as active particle detector. The angular resolving power, which is achieved due to an array of entrance pinholes, can be simply adjusted by modifying the geometry of the experiment and/or the pinhole array itself. The analysis procedure allows for different ion traces to cross on the detector plane, which greatly enhances the flexibility and capabilities of the detector. A full characterization of the TP magnetic field is implemented into a relativistic code developed for the trajectory calculation of each pinhole beamlet. We describe the first test of the spectrometer at the 1PW VEGA 3 laser facility at CLPU, Salamanca (Spain), where up to 15MeV protons and carbon ions from a 3μm laser-irradiated Al foil are detected.

## 1. Introduction

Since the advent of the chirped pulse amplification (CPA) technology [1], the range of accessible intensities on focus for ultra-bright, short-pulse lasers has increased significantly, reaching current values above 10^22^ W/cm^2^ [2]. Such enhancement has paved the way for laser-based particle accelerators, mainly for ions [3,4] and electrons [5]; however, acceleration schemes have been also demonstrated for positrons [6] and neutrons [7].

The range of applications of the accelerated beams is quite rich, profiting from the low-emittance and ultrashort duration (and high peak current) of the generated particle beams, well-fitted characteristics for applications. Specifically, since the demonstration of collimation and monochromatisation of laser-driven multi-MeV ion beams [8,9], their potential employments, such as ultrafast proton probing [10,11,12,13], isochoric heating of dense plasmas [14], fast ignition of inertial confinement fusion reactions [15], material science [16], and medical purposes [17,18], have attained plenty of attention [19].

The compactness and costs of high-power laser facilities are important advantages when compared to conventional radio frequency acceleration facilities, as well as the reduced size of the radio protection requirements. Most of the potential industrial applications of these sources require a high time-averaged particle flux, which stresses the importance of developing high-repetition-rate (HRR) laser sources, targetry instrumentation [20], and diagnostics.

One of the key diagnostics for ion acceleration investigation are the Thomson parabola spectrometers, first developed by Thomson in 1907 [21], i.e., in-line diagnostics which can rate the particles depending on their energy, momentum, and charge-to-mass ratio [22]. The element of the spectrometer sensible to particles can be either a passive detector—for instance, imaging plates (IPs) or a CR39 nuclear track detector [23], which require post-processing to retrieve data, or an active one -microchannel-plate [24]—or plastic scintillators [25], well fitted for HRR operation, due to their ability to perform on-line measurements for every single laser shot. The main drawback of an ordinary TP is the incapability of deconvolving the angular distribution of the measured beam, as only a particular angle of the beam (with an insignificant angular spread) is measured, as the particles measured have to cross a pinhole. This fact also makes this detector specially sensitive to alignment.

Tracing the angular-resolved spectrum of the ions is a vital milestone in the study of the beam properties. For instance, from this kind of work, we have learnt that the most widely used laser-driven ion acceleration mechanism, target normal sheath acceleration (TNSA) [3,4], normally achieved by the ultra-bright laser irradiation of thin metallic films, is able to emit extraordinarily laminar, low-emittance beams from the rear surface of the target [26], coming from a source with a diameter size as big as a few hundred micrometers [10] and a total beam divergence angle around $20^{\circ }$. Potential applications benefit from the transport properties of these laminar beams, which have proven to be suitable when focused to millimetre-sized spots [27,28]. While increasing the resolution of the diagnostics responsible for measuring the ion phase space, new beam features have been discovered, such as the beam pointing deviation from the target normal for certain laser and target conditions [29,30,31]. Tomography-like measurements have also shown that there is a different source size and divergence for each ion energy [32,33,34,35,36]. Non-laminar ion acceleration has also been demonstrated when triggering plasma instabilities under specific circumstances, for instance due to the generation of a preplasma prior to the laser interaction with the target [37,38,39] or the use of ultrathin (nanometric-thick) film targets in the radiation pressure acceleration scheme (RPA) [40,41].

In order to retrieve angular-resolved spectral information about the beam, radiochromic film (RCF) or scintillator stacks are practical diagnostic tools [25,42], typically yielding a discretized spectrum of ΔE≈1MeV, which is much coarser when compared to the continuous spectral TP resolution [22]. Moreover, these diagnostics cannot discriminate between different q/m ionic species. This fact is compensated in some experimental layouts by the combination of perforated RCF stacks and TPs [43]; thus, part of the beam reaches the latter. Despite yielding complementary information, this method is limited in spectral resolution for most parts of the beam.

In this work, we present a multi-pinhole Thomson parabola spectrometer, which combines sharp spectral/angular precision, besides the ionic species sorting capability. Furthermore, the use of a MCP detector device allows for single-shot HRR acquisition. Section 2 describes the basic operation principle of the detector and depicts its physical properties and parameters. The experimental layout where the detector was tested and the analysed results are presented in Section 3.

## 2. Materials and Methods

### 2.1. Thomson Parabola Design and Operation

The Thomson parabola works according to magnetic and electric sector spectrometer principles. The entrance pinhole selects a beamlet composed by ions with a specific charge-to-mass ratio q/m, with q=Ze. The ion charge is deflected by parallel (or antiparallel) magnetic *B* and electric *E* fields of length $l_{2}$. The initial velocity of the particles is perpendicular to the field lines; thus, the deflection directions of the two fields are mutually orthogonal, allowing species separation (by electric field, in *x*-axis) and energy separation (mostly by magnetic field, in *y*-axis) after some propagation distance $l_{3}$. The ions are measured on a two-dimensional spatially resolved particle-sensitive detector; in our case, a MCP. In the small deflection approximation sin(θ)≈θ, considering perfectly sharp and homogeneous fields and nonrelativistic particle energies, the deviation coordinates at the detector plane caused by the Lorentz force are given by
(1)$$x=\frac{qEl_{2}l_{3}}{2E_{kin}},$$
(2)$$y=\frac{qBl_{2}l_{3}}{\sqrt{2mE_{kin}}},$$
where $E_{kin}$ is the kinetic energy of the ion. When combinating (Equation 1) and (Equation 2), we obtain the parabolic equation
(3)$$y^{2}=\frac{q}{m}\frac{B^{2}l_{2}l_{3}}{E}x.$$

Ions with the same charge-to-mass ratio will reach the same parabolic trace on the detector plane; meanwhile, their position along the trace will define their energy. Photons are not deflected by the fields and travel straightly through the source–pinhole axis and impact the MCP, providing the zero deflection reference (used for spectrum data interpretation).

The ultimate energy resolution of the TP depends on the spatial separation of the different energies at the active area of the detector (which depends on the magnetic field strength *B*, its length $l_{2}$ and the distance to MCP $l_{3}$) and on the capability of the system to resolve this separation (which depends on the magnification of the imaging system collecting MCP signal and on the trace thickness δ). In the considered approximation, δ (which is inversely proportional to the spectrometer resolution) is given by the setup geometry and the pinhole diameter *d*, similarly as in a pinhole camera, as $\delta =d+\left(s+d\right)L^{\prime }/L$, where *L* is the distance from source to pinhole, $L^{\prime }=l_{1}+l_{2}+l_{3}$ is the distance from pinhole to detector, and *s* is the source size [44].

Several concept modifications have been proposed to improve the basic functioning of TPs, such as a tunable magnetic dipole [45] or electromagnets [24] for adaptable energy resolution, exotic electrode geometry (trapezoidal or wedged) [45,46,47] for extended retrieval of lower part of the spectrum, transient electric field for time-gated measurement of the beam [48], designs with two in-line entrance pinholes for spatially resolved measurements of the ion source [36], or simultaneous measurements of ion and electron [49] or plasma-emitted extreme ultraviolet radiation spectra [50].

### 2.2. Multi-Pinhole Thomson Parabola Spectrometer

Here, we propose a modification of the basic TP design, consisting on the substitution of the pinhole by a horizontal array of pinholes. This array chops the incoming cone of particles in several beamlets which are simultaneously detected. In this way, we can measure the different angles adding angularly resolved spectral information. Similar strategies were already proposed [32,33,34,35,36,51,52,53,54,55] in most of the cases, dismissing the electric field for charge-to-mass ratio inspections, as the authors claim to accelerate a single ion species (protons). Some references [56] showed the possibility of joining different ion diagnostics in order to have extended information about the beam, but lacking ion discrimination capability at different subtended angles. A few works have proposed [45] or demonstrated [57] an absolute capability of angular spectral-q/m resolution, but with strong limitations in the available species to be investigated, as well as their subtended angles, due to data analysis constraints. In order to facilitate the spectrum retrieval, they managed to avoid crossing traces from neighbouring beamlets at the detector plane.

We propose a more general multi-pinhole TP spectrometer, including the use of electric and magnetic fields for identifying different q/m ions, angular selection of beamlets, and a more generic post-processing method, which does not limit the available ion species to be investigated.

In the case shown here, three pinholes of d=200μm, separated by a=3mm and aligned along the *x*-axis (electric deflection direction, see Figure 1), were drilled into a W 1mm-thick 25mm-diameter plate. A copper nosepiece was designed to easily exchange between these substrates, with different pinhole array combinations with pre-set alignment orientations. A B=0.4T, $l_{2}=75\;\mathrm{mm}$ long permanent dipole magnet is located $l_{1}=13.5\;\mathrm{mm}$ after the pinholes. The direction of the magnetic field (−x) deflects the ions upwards (+y). Magnets are attached to an iron yoke, leaving a gap between poles of 16mm, on-axis with respect to the central pinhole. Two thin copper electrodes are placed over the magnet poles. A variable voltage difference can be applied between electrodes, up to U=10kV, which deviates the particles parallel to the field lines. After a propagation distance $l_{3}=135\;\mathrm{mm}$ a $80\times 30\;{\mathrm{mm}}^{2}$, a single-stage MCP with attached phosphor screen (Hamamatsu F283-12P, 12μm channel diameter) converts the two-dimensional ion traces into visible photons at the rear side of the TP. This signal is acquired by a properly calibrated image recording system, in order to retrieve particle data for every laser shot. All the components are light-tight covered by a shielding made of overlapped layers of Al, polyethylene, and Pb, whose goal is to protect the MCP from secondary radiation sources. The full assembly is 10kg heavy and $130\times 255\times 175\;{\mathrm{mm}}^{3}$ in volume, which makes it relatively easy to set and align inside an experimental vacuum chamber.

A 3D numerical solver was developed for trajectory simulation to provide expected traces in MCP and energy–position relation for all the required charged species. It consisted of a second-order Verlet algorithm and the numerical error was estimated by comparison to analytical results on 0.23% at 20MeV. The magnetic field was mapped by a Hall probe in the central vertical plane (between dipole magnets) and implemented in the code for realistic trajectory calculation (see Figure 2). The electric field could not be measured; therefore, it is implemented in the code as a perfectly sharp and homogeneous field between plates, ignoring edge effects. The numerical simulation provided precise energy position calibrations, accounting for the exact trajectory of particles coming from each pinhole of the array.

In the presented design, proton energies between 300keV and 25MeV are accessible. The ultimate energy resolution of the TP spectrometer is determined by the energy separation at the MCP position (ΔE/Δx) and the trace width. For instance, at 20MeV, ΔE/Δx=4.33MeV/mm, with a measured trace width of 0.30mm, the minimum uncertainty is 4.33MeV/mm×0.3mm=1.3MeV.

In this geometrical configuration, the crossing of traces from different pinholes are abundant and need to be processed (see Figure 3). Crossings produce peak artifacts in the spectrum readout due to intersections. To avoid the artifacts, we proceeded in identifying the species involved on each crossing event. If one works in the pixel position vs the pixel value space, the peaks produced by intersections are aligned in position for all the involved species and can be treated in the same basis. The intersection peaks were identified, located, and eliminated from the raw data to later perform a linear interpolation between both edges of the gap left by extraction. All the intersections were identified, but only the ones producing significant distortion were eliminated. We eliminated the peaks that were still observable after performing a Gaussian smoothing over the data in the pixel space. The convolved Gaussian radius was set to 10 points as Fourier analysis of raw data revealed peaks of size structures from 5 to 20 points. In this way, both statistical variations and small structures up to 20 points were considered as noise and contribute to the root mean square deviation (RMSD) value. The interpolated segments were considered to follow the trend within variations according to this RMSD.

## 3. Experimental Test and Results

The detector was tested at the VEGA 3 petawatt laser facility at Centro de Láseres Pulsados (CLPU, Spain). The target, a 3μm-thick Al planar foil, was irradiated at $10^{\circ }$ to form the normal target in the horizontal plane by the VEGA 3 laser. For the case shown, a single 0.8μm linearly p-polarized pulse of 10.9J (on target) and a 180fs full width at a half-maximum (FWHM) duration was focalized by an F/11 off-axis parabolic mirror onto a 9.7μm FWHM spot, containing 20% of the pulse energy, yielding an averaged intensity of $1.7\times 10^{19}\;W/{\mathrm{cm}}^{2}$ inside the FWHM and resulting in the acceleration of ions from the contamination layer on the target rear surface, rich in H and C atoms, by the TNSA mechanism. As a side effect, ultraviolet and X-rays photons are generated during the laser–plasma interaction, which can define the non-deflected axis of the detector.

The TP spectrometer was carefully aligned so the central pinhole was exactly facing the interaction point in the normal direction of the target surface at a distance L=508mm. The electrode plates were fed with a voltage difference of U=10,000V. The photons from the phosphor screen were collected by an imaging system consisting of an objective (Nikon AF-S NIKKOR 18–105 mm 1:3.5–5.6G ED) with adaptable focal length range of 18–105 mm and a Blackfly PGE-23S6M-C CMOS optical camera of 1920×1200 pixels with a laterial size of 5.86μm. The system was set up to an ultimate magnification of 0.0854 at the detector plane (14.57pixels/mm), which was calibrated thanks to a laser machined pattern in the object plane, next to the MCP. For the detection distance *L*, the choice of pinhole separation a=3mm keeps a reasonably separation of the three traces group, occupying as much as possible the MCP surface. Figure 3 shows the raw traces acquired in a single laser shot.

It is important to note that the source-to-detector distance *L* in the setup prepared is much larger than the source size *s* (typically a few hundred of micrometers in diameter). The magnification of the pinhole camera effect on the detector plane is barely enough to distinguish spatial effects of the source, therefore considering a point-like source for analysis means. The examined angles in this case are 0 and ±α, being $\alpha =\tan^{-1}\left(a/L\right)=0.3^{\circ }$.

A single beamlet proton spectrum is plotted in Figure 4, together with the traces intersections signal peaks, which are considered measurement artifacts and removed. Figure 5 shows the three spectra retrieved from the three analysed proton traces. As expected, due to the small difference of the angles probed when compared to the typical TNSA beam divergence (around $20^{\circ }$), all spectra are similar. This is also true for the carbon ions (see Figure 6). For sake of comparison, different ion species from the same pinhole are plotted in Figure 7. The RMSD was calculated from raw data with respect to smoothed data and some example values are 8310 (a.u.) for left trace protons and 655 (a.u.) for $C^{4+}$ left.

## 4. Discussion

We designed, built, and tested a Thomson parabola which can measure the spectra of discretized beamlets with different emission angles from a laser–plasma interaction experiment at a high repetition rate, simultaneously sorting the ion species by their charge-to-mass ratio.

A novel analysis method, which can examine the crossing parabolic traces on the detector plane, grants access to several variables parallelly. The peak artifacts due to trace crossing are corrected by subtraction of the contribution of the non-desired traces by means of a fitted gaussian intersection function in the pixel position space, ensuring the graph continuity of the analyzed trace.

Future improvements of the detector are considered. An absolute charge calibration can be used a next step for detector refinement, by combining the use of the MCP with CR39 or RCF stacks [24], or possibly by testing the TP in a charge-controlled electrostatic particle accelerator. From the results presented, it also became clear that one of the limitations of the system is the parallel configuration of the electric plates. The detector can be improved by adapting it to a wedged geometry (separation between the electrodes increases with ion path) [45,46,47], thus preventing low energy ions from colliding with the plates, specially pronounced for the ions from the less favourable (left) beamlet (see for instance ”left” carbons spectra in Figure 6 and Figure 7), as also demonstrated in Figure 3. Additionally, an increased size of MCP (or use of other larger 2-D active detector) can be beneficial. The conversion of the TP into an ion wide angle spectrometer [55] (iWASP) is possible by switching off the electric potential of the electrodes, maintaining only the magnetic deflection, and replacing the pinhole array by an horizontal micrometer-wide slit.

Finally, new pinhole array combinations are foreseen to explore different and more numerous probing angles. A larger pinhole imaging magnification can be implemented by placing the detector closer to the target [30], followed by a proper dimensioning of the pinhole array configuration, implying clearer variations detected between traces. For instance, a line of three pinholes with separation a=500μm can set the TP as close as L=12.5mm by avoiding particle collisions with the electrodes, yielding probing angles of 0 and $\pm \alpha \approx \pm 4.5^{\circ }$, where the differences in spectra among the beamlets will be presumably noticeable. Moreover, each pinhole can apply an imaging magnification of $L^{\prime }/L\approx 18$, which will considerably enhance the spatial resolution. Considering high laminar beams, this high magnification imaging mode should be sufficient to resolve changes on the target emission coordinates and beam pointing variations as a function of energy for each beamlet [30,33,34,35], therefore granting a beam emittance characterization. This increased magnification can cause a cost of loss of energy resolution. An estimation of the resolution in the worst case scenario of a pure non-laminar beam provides a value of ±3.5MeV at 15MeV. However, a large percentage of laminarity is expected at these energies to reduce the trace thickness and increase the resolution. In any case, the observation of how laminarity depends on energy is itself an important achievement, and this type of setup can provide a direct measurement of it. On the other hand, it seems possible to apply the multi-pinhole TP to diagnose energy (and species) laser-generated ion beamlines. In such a case, the detector field configuration can be adapted in order to comprise an adequate range of energies, therefore improving the energy resolution considerably. All these possibilities show the potential interest of such kind of detectors for beam analysis of novel acceleration mechanisms under research, such as collisionless shock acceleration (CSA) [58] or RPA [40,41,59], as well as for measurements of transported ion beamlines, well fitted for applications.

## Figures and Tables

**Figure 1 sensors-22-03239-f001:**
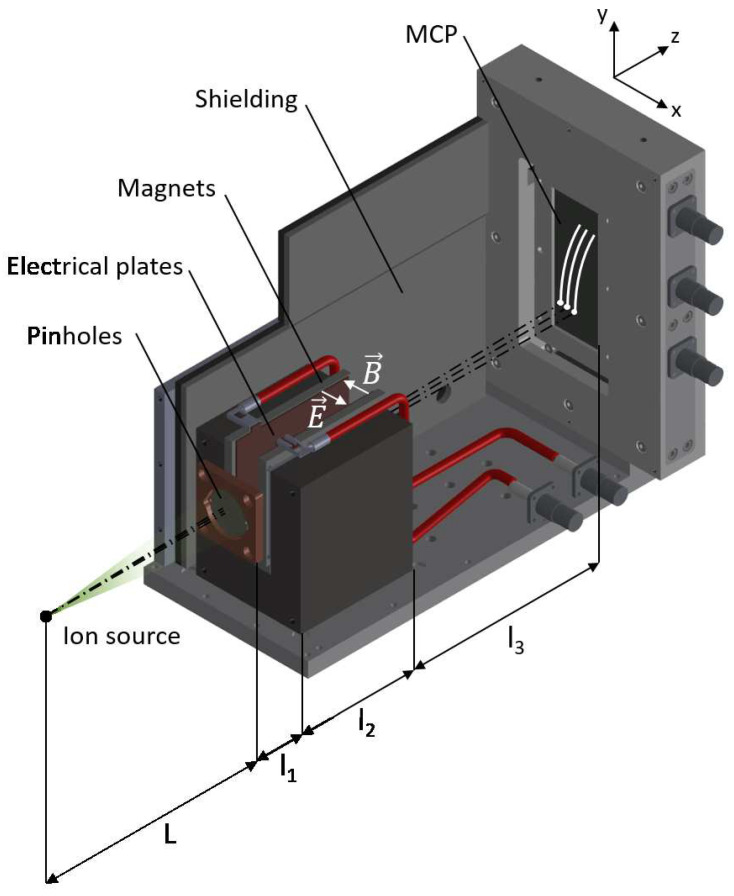
Multi-pinhole TP spectrometer design.

**Figure 2 sensors-22-03239-f002:**
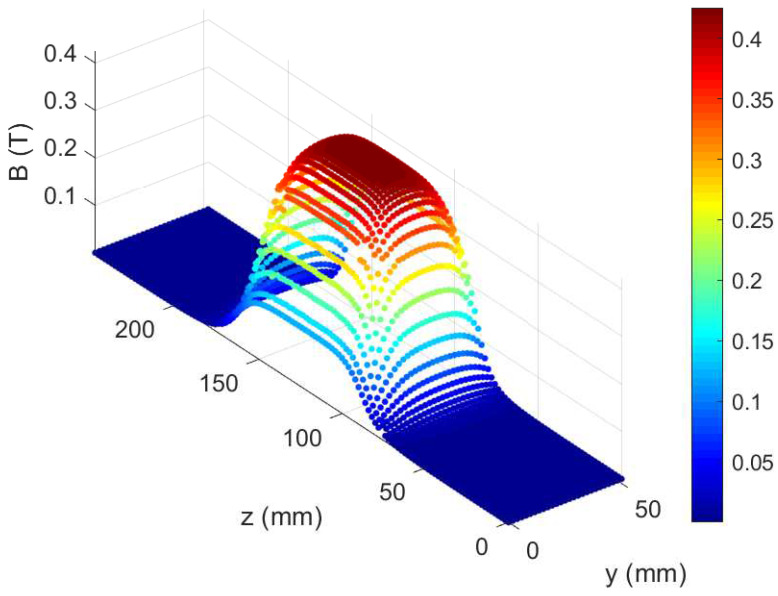
Measured magnetic field distribution. Ions propagate in z-direction through the field towards the MCP, deflected on the y-direction by the Lorentz force.

**Figure 3 sensors-22-03239-f003:**
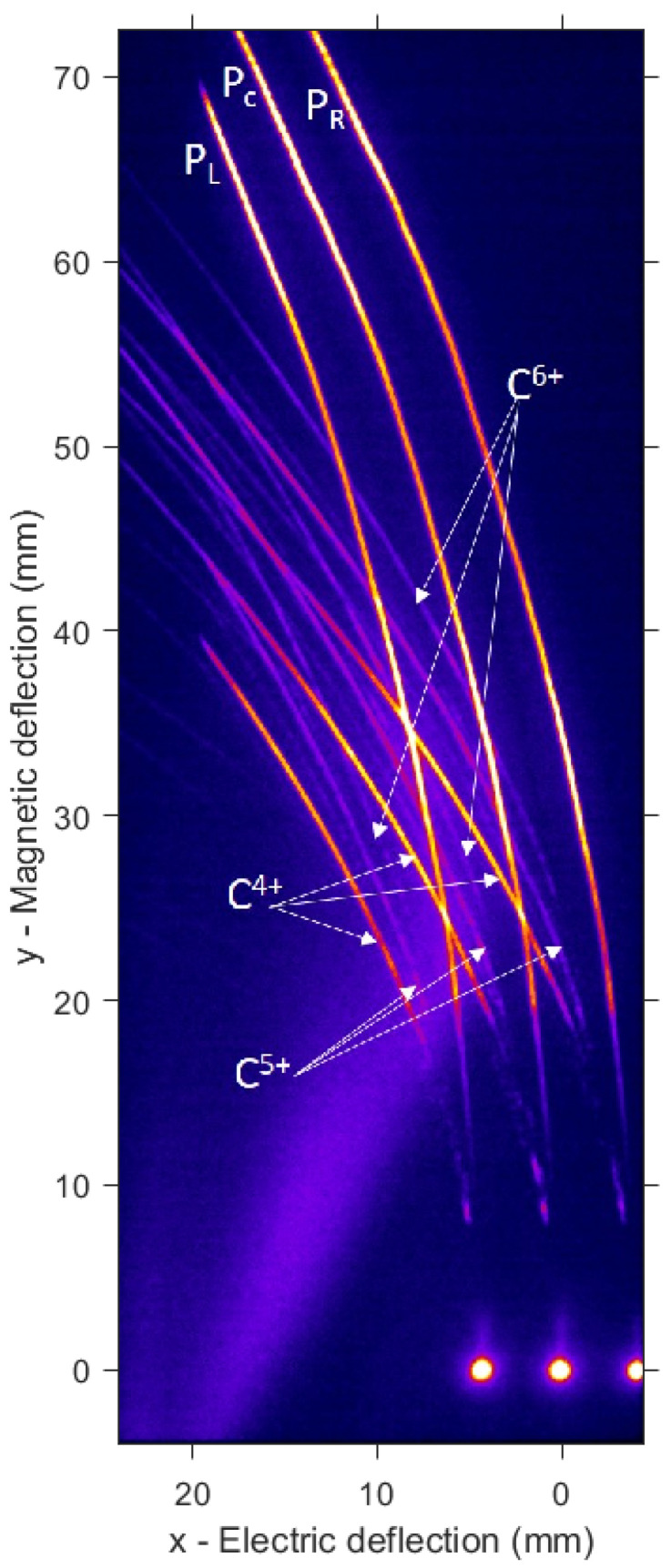
Multi-pinhole Thomson parabola traces obtained from a single laser shot at VEGA 3. The origin of the coordinate system corresponds to the zero-deflection point of the central beamlet. $P_{L}$, $P_{C}$, and $P_{R}$ corresponds to the tracks related to the left, central, and right beamlets, respectively. $C^{n+}$ indicates the three n-charged ion traces. The halo at the bottom left corner is originated by a leak of white light from plasma emission, which reaches the MCP through the junction of MCP structure and shielding.

**Figure 4 sensors-22-03239-f004:**
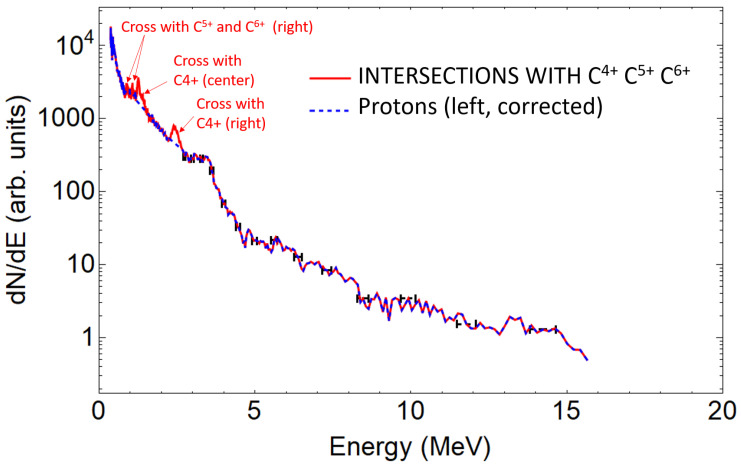
Left trace proton spectrum in logarithmic scale. Blue: corrected spectrum. Red: same spectrum showing the peaks subtracted, corresponding to trace crosses. Several representative error bars are plotted, picturing the estimated energy resolution.

**Figure 5 sensors-22-03239-f005:**
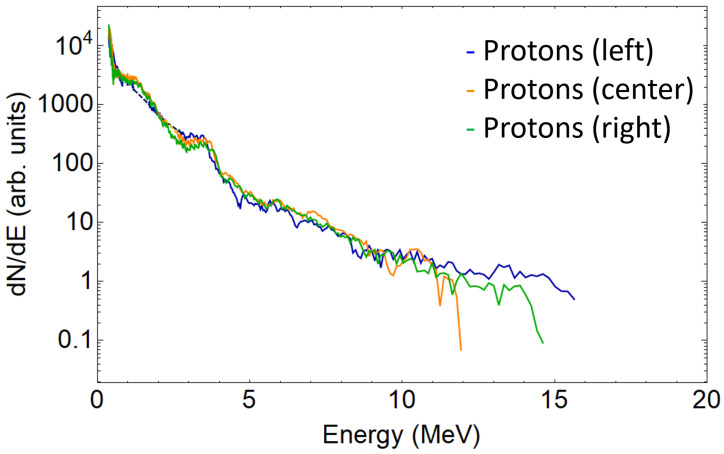
Left, center, and right proton beamlets spectra in the logarithmic scale. Dashed lines show the reconstructed spectrum range. It is important to note that the crosstalk between traces lays at different energy locations for each beamlet. This effect is clear from Figure 3; the relative horizontal position of each proton trace makes the crosses appear at different energetic levels. Note for instance that the proton right trace does not suffer any crosstalk; thus, no correction is applied.

**Figure 6 sensors-22-03239-f006:**
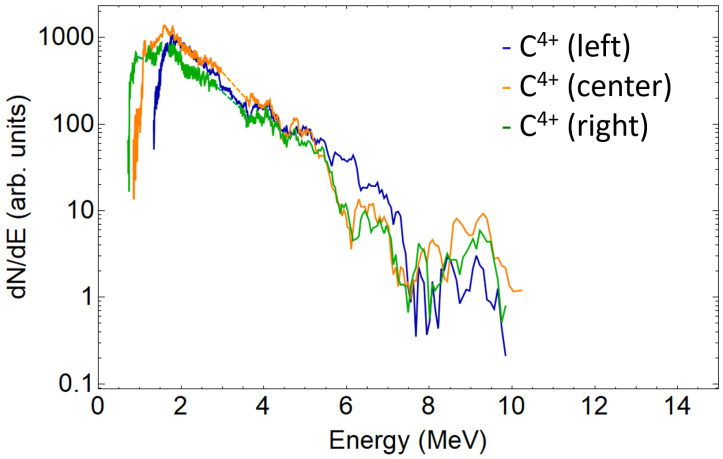
Left, center, and right $C^{4+}$ beamlets spectra in the logarithmic scale. Dashed lines show the reconstructed spectrum range.

**Figure 7 sensors-22-03239-f007:**
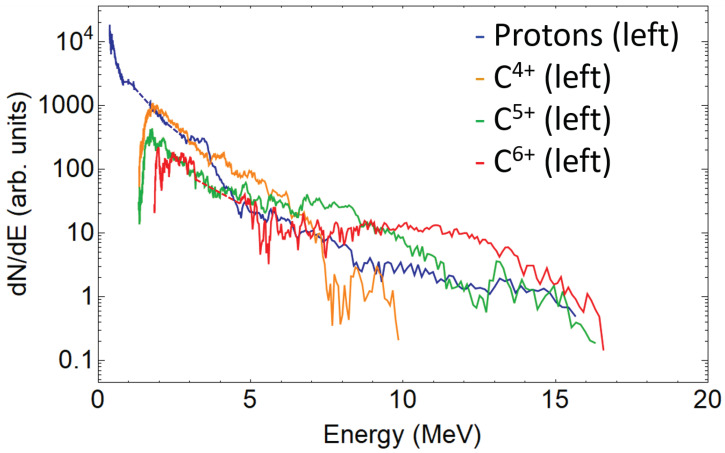
Left beamlets spectra for protons and $C^{4+}$, $C^{5+}$, and $C^{6+}$ ions in the logarithmic scale. Dashed lines show the reconstructed spectrum range.

## Data Availability

Not applicable.

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
