# Peer review of "Angular-Resolved Thomson Parabola Spectrometer for Laser-Driven Ion Accelerators"

_sensors, 2022, doi:10.3390/s22093239_

Round 1

Reviewer 1 Report

This work presented by Lopez et al, provides a possibility of  small size charge/mass separation devices. The overall presentation is good. The work is publishable but few points need to be clarified.

Minor edit

  1. Page 4, line 127; Authors mention the pinhole size and distance between them but there is no justification for this choice either experimental or theoretical.
  2. Continuing from the previous question on spacing of pinholes and their size, the crosstalk issue is present. What are the steps need to be taken to minimize the interference?
  3. Page 5, first paragraph; Energy separation at the MCP position 4.33 MeV/mm was obtained theoretically or experimentally?
  4. Figure 3, the curves/traces from same charge does not look originating from the same curvature of radii. Is it due to the slight difference in the original position/angle at the MCP?
  5. Figure 4, the left, right and center trace spectrum has crosstalk at different energy locations? What is the reason behind this and it will be helpful if the explanation can be included in the text.
  6. Figure 7, are the protons/carbon traces are collected together or separately? Also, how the crosstalk different for protons and carbon?

Author Response

Please se the attachment.

Reviewer 2 Report

The goals and the methods are clear and the paper is well organized. Nevertheless, please the authors consider the following suggestions/corrections: 

Line 3: High repetition rate (HHR) compatibility is granted --> the authors mean "the compatibility is guaranteed"?

line 30:  require of a --> require a 

line 54: responsible of measuring --> responsible for measuring

line 69: most part --> most parts of the beam

Fig. 1. E and B in the figure are not clear. Should not them be orthogonal? i.e. Should B directed vertically? From their direction they look antiparallel. As far as I understand, figure 1 represents your particular experiment with multiple pinholes as demonstrated by the imaged tracks, while at line 81 the figure is cited to desctibe the general working principle. It would be more clear if you would have put here a general picture of the principle and later showing your 3 pinholes TP (as correctly cite this figure at line 128).

line 81 fig.1 --> fig. 1. But I would  use all along the text the word "figure"

line 94-96. What is the magnetic spatial resolution? Can the authors define it? Furthermore the sentence can be clairified: do the authors mean that the energy resolution depends on the magnetic spatial separation , on the magnification and on the particle trace thickness?

line 112: proposed in most of the cases

line 121: constains --> constraints

line 121: is not clear to me the sentence "ion traces from different beamlets are not allowed cross at the detector plane". Traces cannot cross on the detector plane for some physical reason or  are they not allowed crossing for some practical reason? 

line 142:  to protected --> to protect

line 146: it consisted on a --> it consisted of a 

Line: 151: as as  --> as a

Line 176. Is not stressed enough that the accelerated particles are carbon. How are they obtained from al Aluminum foil? Several other particles with different Z emerges from the foil? Is this the reason of several tracks to be subtracted? Can the authors describe this in the text?

In order for me to  understand better:  am I right if I say that the laser beam has the double function of being source of photons for zero position/no deflection on the detector and for source of ions? If yes please specify in the text for the non expert reader. 

Line: 194: tipically few --> tipically a few 

Line: 198 with --> which are

FIgure 3. The caption "PL, PC and PR point to the left, central and right proton traces respectively" means: PL PC PR corresponds to the tracks related to the left, central and right beamlets, respectively ? Please rewrite the caption more clearly.

Fig. 3. Do you have any idea of what generate that halo starting at x = 20, y = 0 and going diagonally towards the center of the image? At least in the caption the authors could mention it and explain its presence, if known.

Line 207: with different emission angle --> with different emission angles

Line 213: corrected by substraction --> corrected by subtraction

Line 237: will enhance considerably the spatial --> will considerably enhance the spatial

Line 243: However, large percentage --> However, a large percentage

Line 247: In such case, --> In such a case,

Line 249: improving therefore the energy --> therefore improving  the energy

General comment 1: along the text sometime the reference is given without blank space, e.g. at line 108 "spectra[50]" and sometime with space e.g. at line 107 "electron [49]". I suggest to go through the text and include blank space everywhere is needed. 

General comment 2.  is there any real picture showing the device? If yes it could be nice to see it coupled with the CAD drawing (fig. 1).

Author Response

Please se the attachment.
